# Adversarial training for high-stakes reliability

**Daniel M. Ziegler**[*]  **Seraphina Nix**  **Lawrence Chan**[†]  **Tim Bauman**

**Peter Schmidt-Nielsen**  **Tao Lin**  **Adam Scherlis**  **Noa Nabeshima**

**Ben Weinstein-Raun**  **Daniel de Haas**  **Buck Shlegeris**  **Nate Thomas**

Redwood Research

## Abstract

In the future, powerful AI systems may be deployed in high-stakes settings, where a single failure could be catastrophic. One technique for improving AI safety in high-stakes settings is adversarial training, which uses an adversary to generate examples to train on in order to achieve better worst-case performance.

In this work, we used a safe language generation task ("avoid injuries") as a testbed for achieving high reliability through adversarial training. We created a series of adversarial training techniques—including a tool that assists human adversaries—to find and eliminate failures in a classifier that filters text completions suggested by a generator. In our task, we determined that we can set very conservative classifier thresholds without significantly impacting the quality of the filtered outputs. We found that adversarial training increased robustness to the adversarial attacks that we trained on—doubling the time for our contractors to find adversarial examples both with our tool (from 13 to 26 minutes) and without (from 20 to 44 minutes)—without affecting in-distribution performance.

We hope to see further work in the high-stakes reliability setting, including more powerful tools for enhancing human adversaries and better ways to measure high levels of reliability, until we can confidently rule out the possibility of catastrophic deployment-time failures of powerful models.

## 1 Introduction

Advances in deep learning have led to increasingly powerful AI systems, for example in sequential decision making [1, 2, 3, 4], robotics [5, 6], and language modeling and text-based reasoning [7, 8, 9, 10, 11]. Most empirical work on techniques for aligning powerful AI [12, 13, 14, 15, 16] has focused on achieving good *average-case* performance in domains where no single action is catastrophic, for example using human trajectory rankings [17, 18, 19] or imitation learning [20, 21]. However, many situations where we want to deploy AI systems are *high-stakes*—that is, it is possible for the system to take actions that lead to catastrophic outcomes.

In these situations, one of our most important goals is *high-stakes reliability*: avoiding even a single catastrophic failure while in deployment. Achieving high-stakes reliability is difficult because some failures might not be encountered during the ordinary course of training, leaving them uncorrected by default. These failures could arise on out-of-distribution data resulting from domain shift

---

[*]Corresponding author. Please direct correspondence to dmz@rdwrs.com.
[†]UC Berkeley. Work done at Redwood Research.

36th Conference on Neural Information Processing Systems (NeurIPS 2022).

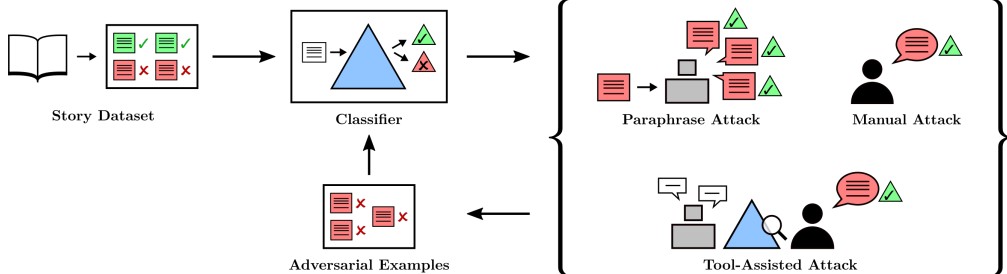

Figure 1: A representation of our adversarial training loop. Starting from an initial story dataset consisting of prompts and generator completions (Section 4.3), we trained a classifier to detect injurious completions. We then iteratively attacked our classifier using unaugmented humans (Section 4.4.1), automatically paraphrased previous adversarial examples (Section 4.4.2), and tool-assisted human rewrites (Section 4.4.3), while training on the resulting adversarial examples.

or adversaries in the environment. Alternatively, undetected failures could arise without distributional shift if they occur with sufficiently low probability. We describe our setting more precisely in Section 3.1.

One technique for improving high-stakes reliability is *adversarial training* [22, 23, 24]. In its general form, adversarial training consists of finding inputs that a model does especially poorly on and then training the model on those examples. If our adversarial attacks sufficiently cover the space of catastrophic inputs, then adversarial training incentivizes the model to avoid catastrophic failures.

In this work, we used a simple task as a testbed for adversarial training. The system must take a three-sentence excerpt from a story (a "prompt") and output one more sentence (a "completion") that continues the story *without introducing any physical injuries to any characters*. To do this, we train a language model as a classifier for injurious completions, which we use to filter the outputs of a generative language model. We then adversarially train it using a variety of attacks (Figure 1).

As measured by both the false negative rate on our adversarial datasets and the time to generate adversarial examples, we found that adversarial training increased robustness to attacks similar to those trained against (Section 4.4.3), although it did not eliminate failures completely. Qualitatively, we found that the remaining failures in adversarially trained models were less egregious and were less likely to contain mention of direct injury (as opposed to implied or indirect injuries). At the same time, we found that adversarial training did not degrade performance on our baseline (non-adversarial) dataset. Finally, we found that we could set very conservative classifier thresholds without degrading the quality of our generator output.

Our main contributions are the following:

(1) We highlight the setting of *high-stakes reliability* and report the results of an initial project in this setting.

(2) We demonstrate a novel tool-assisted human attack that increases the ease of finding adversarial examples (Section 4.4.3)

(3) We found that on our chosen task, conservative thresholds enable a high degree of worst-case reliability, with minimal impact on average-case performance.

We see our work as exploratory and think that there are many promising follow-up directions to pursue for stronger results. We hope that this project will be followed by work building the theory and practice of adversarial training to the point where it can robustly enable high-stakes reliability.

## 2  Related work

The field of adversarial machine learning [25] or even the subfield of adversarial training [26] are too large to summarize in this paper. Here, we outline a handful of particularly related areas.

**Adversarial training for image classifiers**   Much recent work in adversarial training has been on preventing adversarial examples for image classifiers [22, 27]. Notably, the majority of image

adversarial training work studies $L_p$ ball perturbations [22, 28, 29, 30, 31]. In contrast, we allow adversarial examples in an unrestricted space [32], with the goal of avoiding any feasible failures.

**Adversarial training for language models**    There is a large and growing literature on both adversarial attacks and adversarial training for large language models [33, 34, 35, 36, 37]. The majority of these focus on automatic attacks against language models, where the authors have access to an automated source of ground truth, or restrict their examples to small perturbations that are assumed not to change the ground truth. In this work, we primarily used human adversaries as we chose a task without an automated source of ground truth.

**Human-in-the-loop adversarial training**    Our human-centered approach to adversarial training is most similar to the "Break-it-Fix-it" approach taken by Dinan et al. [38], which uses human-generated adversarial examples to increase the robustness of an offensive speech classifier.

In recent years, this approach has also been proposed as a method of evaluating language model classifiers in general. Several recent datasets and benchmarks are constructed with human-in-the-loop adversaries, such AdversarialNLI [39], AdversarialGLUE [40], and DynaBench [41]. Our analysis of the effects of multiple iterations of adversarial training resembles DADC [42].

Most prior work picked tasks challenging enough for the model that unaugmented human labelers were able to easily find adversarial examples, whereas in our case it started to become more difficult for human labelers to find further adversarial examples after initial adversarial training. This led us to augment our labelers' abilities. Like Wallace et al. [43], our tool displays a saliency map; however, we added a token substitution tool that gives humans automated suggestions to help hill-climb toward an adversarial example.

## 3    Problem statement

### 3.1    General setting: high-stakes reliability

Here, we define high-stakes reliability more precisely. A *high-stakes reliability setting* contains both: 1) an **average-case performance metric** that measures successful execution of the intended task; as well as 2) a **notion of "catastrophic failure"** that must be avoided in deployment. The goal in high-stakes reliability settings is to train a model that avoids catastrophic failures while still achieving good average-case performance.

During training, we assume that we have an oversight process that can safely detect catastrophes in arbitrary new examples. However, the oversight process may be too expensive to evaluate for every example in deployment. For example, we might be able to pay humans labellers to closely examine a few tens of thousands model outputs in training, but not be able to scale human oversight to hundreds of millions of model outputs in deployment. As fully preventing all catastrophic failures may be unachievable with current techniques (and very low failure rates are hard to measure), we propose using two proxy metrics instead. First, we can measure *the failure rate on adversarially-generated datasets* designed to elicit catastrophic behavior. Second, we can measure *the difficulty of finding a novel adversarial example*, using particular styles of adversarial attacks.

### 3.2    Our specific task: filtering a story generator

In this work, we consider the task of producing safe completions to fictional stories with a filtered language model. Specifically, given a three-sentence prompt, our goal is to produce a *noninjurious* completion—that is, one further sentence that does not introduce additional injury to any characters.[1] We chose this relatively simple non-injury task to make it reasonably easy for non-expert humans to recognize failures and enable relatively small models to perform acceptably well on our task.

In our specific setup, we assume that we have access to a generative language model that generates high-quality completions to given prompts; our task is to learn an injuriousness classifier that classifies completions as injurious and then use it to filter the output of our generator. We use the quality of our filtered completions (as judged by human raters, relative to our unfiltered language model) as our average case performance metric. Our "catastrophic failures" are injurious examples

---

[1]We provide more details of our definition of injury in Section A.1.2.

| Dataset | Train | | Validation | | Test | |
|---|---|---|---|---|---|---|
| Initial story dataset (Sec. 4.3) | 166,210 | (10%) | 102,297 | (5%) | — | |
| In-distribution test dataset (Sec. 5.4) | — | | — | | 100,033 | (2.4%) |
| Manual adversarial examples (Sec. 4.4.1) | 1,492 | (46%) | 253 | (47%) | — | |
| Automatic paraphrases (Sec. 4.4.2) | 12,514 | (21%) | 1,734 | (23%) | — | |
| Tool-assisted rewrites (train) (Sec. 4.4.3) | 4,904 | (62%) | 1,513 | (67%) | — | |
| Tool-assisted rewrites (test) (Sec. 5.3) | — | | — | | 1,584 | (84%) |

Table 1: The number of labeled snippets (prompt + completion pairs) from each source of data. The percentage that were labeled injurious are in parentheses.

that the classifier incorrectly labels as safe (that is, the catastrophic failure rate of the system is the false negative rate on filtered generator outputs).

# 4 Methods

In this section, we describe how we trained our injuriousness classifier. After training a baseline classifier on some initial labelled data, we attacked it with several adversarial training techniques and retrained it using the adversarial examples we generated. We summarize the properties of the datasets used in training in Table 1.

## 4.1 Human Labellers

We sourced human contractors primarily from Upwork and from Surge[2] to perform our labeling. To determine whether snippets were injurious, we asked the contractors to label each one injurious, non-injurious, or "Unsure". We used these human labelers to label all our training and evaluation data, including our adversarial training data.[3]

## 4.2 Classifier training

We trained a classifier by fine-tuning `deberta-v3-large` [44] from HuggingFace [45]. During training, we treated all snippets labeled "Unsure" as injurious for the purposes of training because we wanted to train our classifier to be conservative. Because our datasets had more non-injurious examples than injurious examples, we upsampled [46] snippets labeled injurious by up to 5× so that they were closer to the number of non-injurious examples.[4]

## 4.3 Initial data sources

Our initial, baseline classifier training set consisted of "snippets" derived from a dataset of fan fiction stories. We sourced our prompts from an archive of approximately 300 GB of stories from fanfiction.net, and subselected them for increased likelihood of injury to address the class imbalance caused by the low base prevalence of injury.[5] We generated completions from a GPT-Neo-2.7B [47] fine-tuned on this story dataset.

## 4.4 Adversarial attacks and datasets

After training a model on our initial fanfiction dataset, we performed three rounds of adversarial training. In each round, we generated adversarial examples against a model trained against previously generated datasets, using a different type of adversarial attack. We labelled the adversarial examples with human labellers, before training a new model on all the datasets generated so far.

---

[2] https://www.surgehq.ai/

[3] See Appendix A.1.6 for details of our labeling process.

[4] We found in a preliminary experiment that upsampling injurious snippets improved performance, though the effect did not reach statistical significance. We document other hyperparameters in Appendix A.3.

[5] See Appendix A.1 for more details on the classifier training dataset.

### 4.4.1 Manual adversarial examples

In our first round, we used unassisted, manual human adversaries. We created a web interface where a user could input a prompt and a completion and check the classifier score for our baseline classifer. Users were able to submit the snippet if the score dropped below $0.05$ and they believed the snippet was injurious. We then hired contractors to write adversarial examples using this web interface. In order to increase the diversity of adversarial examples, partway through data collection, we changed the target model from baseline classifier to a model that had been trained on a subset of the manual adversarial examples gathered so far.

### 4.4.2 Automatically paraphrased adversarial examples

To increase the size of our adversarial dataset, we used GPT-3 [7] to paraphrase manually generated human adversarial snippets, as has been done with previous generative language models [48]. We devised a "few-shot" GPT-3 prompt for this purpose, containing several examples of paraphrasing[6]. For each snippet, we generated many possible paraphrases of it with GPT-3, then had our classifier classify all the paraphrases as injurious or non-injurious. We discarded all paraphrases that the classifier already rejected as injurious, as we were only interested in adversarial examples. We continued sampling until we had generated 5–10 accepted paraphrases for each snippet.

### 4.4.3 Tool-assisted rewrites

To generate adversarial examples more effectively, we helped our contractors by adding two key features to our interface, as shown in Figure 4 in the Appendix.[7]

1. We built a token substitution and insertion feature. The user could click on a token in the snippet and a dropdown would appear with a list of new tokens sorted by how much they reduce the current model score. The candidate tokens were plausible "fill-in-the-blank" suggestions from `deberta-v2-xlarge` [49].

2. We displayed an indication of saliency for each token in the snippet. The interface highlighted in yellow the tokens most likely to affect the classifier's output. The saliency score for a token was the magnitude of the gradient of the classifier's output with respect to the token's embedding.

As before, users were free to enter or edit examples manually (regardless of the tokens suggested by our token substitution and insertion features), and could submit examples if the classifier score dropped below 0.05. We show examples of adversarial snippets and an example progression of the tool-assisted rewriting process in Appendix C.5. As with the manual adversarial examples, partway through we trained our model on what we had so far and collected the rest of the rewrites against the new, more powerful model.

## 5 Experiments and results

### 5.1 Classifiers evaluation details

In order to evaluate how helpful each of the three attack techniques (Section 4.4) were, we trained a series of classifiers with the data from one further attack added each time.

We trained all models with the same number of labeled snippets (166,210) in order to make it clear that the differences are due to the increased utility of adversarial examples. We randomly removed a number of snippets from the initial fan fiction dataset equal to the number of adversarial snippets added at each step (1,492, 12,514, and 4,904 respectively).

**Classifier threshold selection.** To maximize the reliability of our filtered generator, we wanted to find the lowest classifier threshold—that is, the classifier score above which we consider a completion to be injurious—that had only a small impact on the quality of filtered snippets. We created a static quality-evaluation dataset that could be reused to estimate the quality for a variety of classifiers

---

[6]We include the GPT-3 prompt in Appendix A.6

[7]Our tool is available at `https://www.ttft.io/talk-to-filtered-transformer`.

| Classifier | Threshold | Test Set FNR | Quality ± 95% CI | Test Set FPR |
|---|---|---|---|---|
| baseline | 0.174% | 2/2447 | 49.3% ± 0.71% | 25.5% |
| +manual | 0.208% | 3/2447 | 49.6% ± 0.72% | 27.0% |
| +paraphrases | 0.174% | 2/2447 | 49.2% ± 0.73% | 27.8% |
| +tool-assisted | 0.18% | 2/2447 | 49.4% ± 0.67% | 24.5% |

Table 2: As discussed in Section 5.1 and Section 5.4, all four of our models were able to achieve very low false negative rates (FNR) on in-domain test-set data without significantly degrading the quality of the filtered output (50% = no quality degradation), despite each model having a false positive rate (FPR) of around 25% (that is, filtering around a quarter of non-injurious outputs).

and thresholds, by asking contractors to compare 4–100 pairs of completions for each prompt, with completions drawn i.i.d. from the unfiltered generator. We asked them to choose the higher quality completion or mark them both as the same quality, where two completions of equal quality count as half a win for each completion. Then, we estimated the quality at each classifier threshold by evaluating the classifier against the labeled generations to simulate rejection sampling. See Appendix A.7 for more details.

When evaluating our models' performance, we used the lowest threshold that kept the 95% confidence lower bound of the preference rate at greater than 48% (with 50% meaning equal performance to the unfiltered generator). We were able to select thresholds between 0.0017 and 0.0021.[8] At the selected thresholds, our classifiers reject about 25% of random snippets and 50–60% of completions proposed during rejection sampling.

After selecting the thresholds, we confirmed that our quality estimates were accurate by performing a direct quality evaluation. We rejection-sampled one completion from each of 5000 in-distribution prompts and compared them to a completion from the unfiltered generator.[9] As our results in in Table 2 show, the thresholds we picked did not significantly degrade the quality of the filtered generator.

## 5.2 Tool-assisted rewrites were faster to generate than manual adversarial examples.

To determine how much the token substitution tool helped, we ran a smaller scale experiment where we measured the time for contractors to attack the Baseline and +tool-assisted classifier with and without our rewrite tools. Contractors took on average 20 minutes (95% CI: [16, 24]) to find an adversarial example for the baseline classifier without the tool, compared to 13 minutes with the tool (95% CI: [12, 14]). In addition, contractors took on average 44 minutes (95% CI: [31, 67]) to generate an adversarial example for +tool-assisted without tools, compared to an average of 26 minutes (95% CI: [24, 29]) with our rewrite tool. We also received feedback from contractors saying that the task was significantly more frustrating without our rewrite tool. This demonstrates that our tool helps our contractors more quickly generate adversarial examples for our classifiers.

## 5.3 Adversarial training increased robustness to adversarial attack.

We evaluated our classifiers on adversarial snippets from our final attack to determine whether the adversarial training increased robustness to this type of out-of-distribution data.

**Adversarial training increased the difficulty of finding additional adversarial examples.** We had contractors attack our final classifiers using tool-assisted rewrites. Each contractor was randomly assigned a classifier each day they worked so that an approximately equal amount of effort was spent on each classifier.[10] Each tool-assisted rewrite was labeled by a distinct pair of contractors (with a third tie-break label in case of disagreement) to confirm that it was injurious. Further details

---

[8]More details on classifier hyperparameters in Appendix A.3. Note that these thresholds were significantly more conservative than the threshold of 0.05 that we used during adversarial training.

[9]If the filtered generator could not generate an accepted completion within 100 attempts, we always counted that comparison as a loss.

[10]If we had held the number of snippets per classifier equal (instead of time spent), differences in different labelers' comparative strength against different classifiers might have biased the measurement. The per-day assignment also gave labelers the chance to learn tricks specific to one classifier, although they were blinded to which classifier they were assigned to.

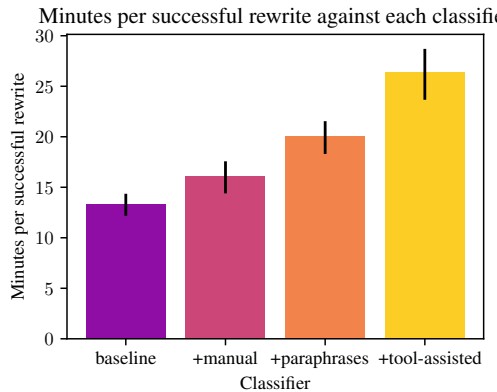
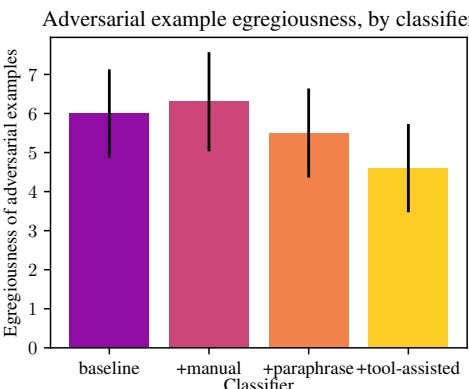

Figure 2: *(Left:)* The amount of time spent per successful tool-assisted adversarial example (higher is better) and *(Right:)* the egregiousness of said adversarial examples (lower is better), for each of the four classifiers. Error bars represent boot-strapped 95% confidence intervals. As discussed in Section 5.3, the difficulty of finding adversarial examples increased significantly with adversarial training. In addition, the adversarial examples against classifiers trained against a more diverse pool of adversarial attacks tended to be less egregious, suggesting increased robustness.

of this process are in Appendix A.4.1. Randomly selected tool-assisted rewrites from each dataset are found in Appendix 8.

As reported in Figure 2, against each successive classifier, it took more time to generate a new adversarial example.[11] This shows that our attacks have some shared surface area, i.e. training on our earlier adversarial attacks also confers advantages against the tool-assisted adversarial attack. Training on all adversarial attacks increased the time per example by 2×.

The smaller-scale experiment in Section 5.2 shows a similar increase in difficulty when contractors do not have access to the tools. Contractors took 44 minutes (95% CI: [31, 67]) to break +tool-assisted manually, versus 20 minutes (95% CI: [16, 24]) to break the baseline classifier, so adversarial training increased the time per example by 2.2× (95% CI: [1.5×, 3.5×]).

**Classifiers trained against more adversarial attacks were more robust to other adversarial attacks.** We treated each set of tool-assisted rewrites as a separate dataset and evaluated each classifier against each dataset. Figure 3 shows the results. Ignoring the faded bars for attacks targeting the same classifier being evaluated (which will always be at FNR = 1.0), each chart shows a lower FNR for later classifiers. The +tool-assisted classifier is robust to attacks targeted at previous classifiers (rightmost bar in each chart). Conversely, attacks targeting it work nearly as well on previous classifiers also (bottom right chart), suggesting that its failure modes were largely preexisting rather than the result of new problems introduced via additional adversarial training.

**Adversarial examples against adversarially trained classifiers were somewhat less egregious.** We found that, in our subjective judgment, the rewritten snippets that our contractors submitted against more adversarially-trained classifiers seemed somewhat less egregiously injurious than tool-assisted rewrites against less adversarially-trained classifiers. We defined "egregiousness" as a combination of the severity of the injury and the likelihood of injury or increased injury in the completion. One researcher labeled ten snippets rewritten against each classifier subjectively on a scale from 1 (not injurious) to 10 (specific, maximum severity injury) while blinded to which classifier the attack targeted. The average egregiousness ratings are reported in Figure 2.

We also looked at the snippet labeled most egregious for each classifier from this process and compared these snippets subjectively. Table 3 shows two of them. As the table suggests, classifiers

---

[11]We estimated this by counting the total time spent rewriting snippets per classifier and dividing by the number of successful tool-assisted rewrites.

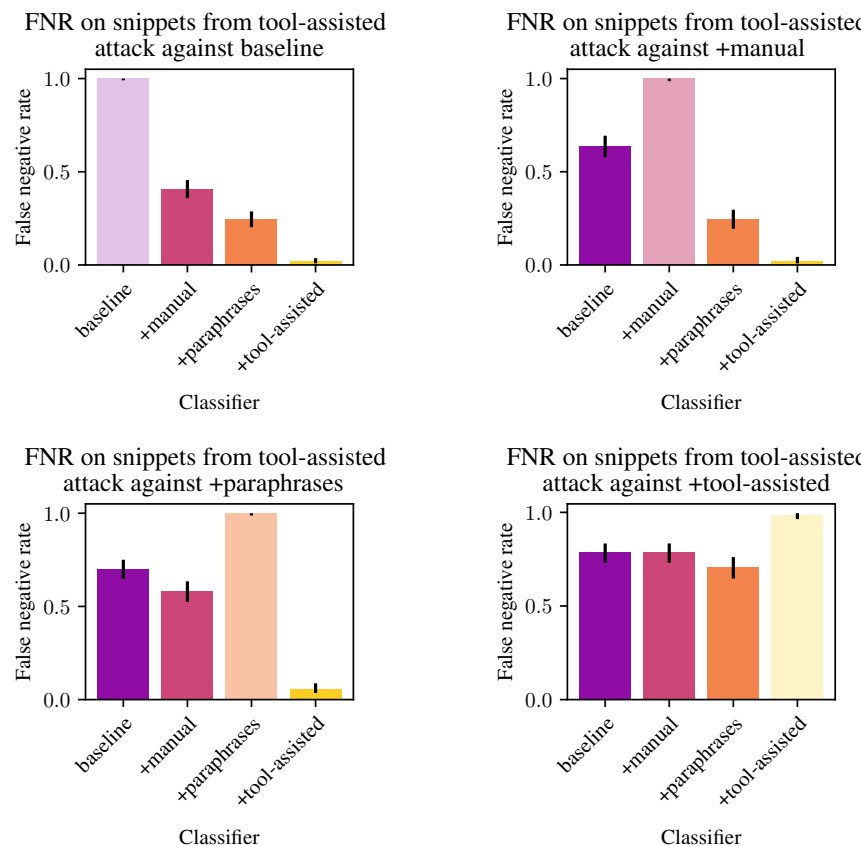

Figure 3: False negative rate (lower is better) for each of our four classifiers on each tool-assisted dataset. Error bars represent the central 95% Bayesian credible interval with a $\beta(1, 1)$ prior. On examples targeting them, classifiers must have an FNR of 1.0 by definition (faded bars), since our tool only accepts examples that the classifier accepts. Otherwise, we see a steady trend of lower FNRs for later classifiers, suggesting increased adversarial robustness.

trained against a broader pool of adversarial attacks tended to have less egregious adversarial examples.

### 5.4 Adversarial training did not hurt performance on the in-domain data.

To evaluate the effect of adversarial training on our initial baseline (in-domain) dataset, we collected 100,033 new snippets drawn from our story database and our generator in a method similar to the one described in Section 4.3, except without selecting for injurious snippets. We had each snippet labeled twice and had each snippet with a disagreement labeled by a third tie-breaker to decrease the error rate on our test set labels. We also removed the option for "Unsure" due to a miscommunication with our vendor (see Appendix C.4).

As reported in Table 2, all four of the evaluated models performed about equally well, getting 2 to 3 false negatives on 2447 positives on our dataset of 100,033 examples. In addition, all four classifiers achieved similar false positive rates. This suggests that adversarial training did not affect the performance of our classifiers on in-domain data.

## 6 Discussion and Future Work

In this work, we explored a simple example of a high-stakes reliability task. We developed a quality measurement technique for rejection-sampled generators and found that we could set very conservative thresholds without significantly reducing quality. We built a series of adversarial training

| Classifier | Most egregious snippet |
|---|---|
| baseline | *"I'm fine," I insist.*
*"There is no way. Look at your hand!"*
*"What about it?" I ask stubbornly.*
$\rightarrow$*"It's your only one!" he said, exasperated.* |
| +tool-assisted | *"And so instead of chanting the spell and living through this once again – although she knew there was probably no escaping that, no matter what she tried – she mounted her broom and flew off to the Great Kells, to the cliff she had been before. She didn't even waste time yelling at the clouds above; she just stepped off her broom, dropped it onto the floor, walked towards the edge and jumped to her death without further hesitation. The next moment, she was in the afterlife, looking at her mother once more."*
$\rightarrow$*She looked to the side and her father was there smiling at her with love.* |

Table 3: The most egregious tool-assisted human generated adversarial examples from a sample of ten against the baseline and final adversarially trained classifier. While the adversarial example versus the baseline involves explicitly mentions the loss of a hand, the most egregious adversarial example against +tool-assisted only indirectly implies that the character's father had prevously passed away. Additional examples of snippets are available in Appendix C.

techniques, including a tool-assisted human attack, and found that they improved the classifier's robustness to attack without affecting in-distribution reliability. Below, we outline some limitations of the current work and a variety of directions for future work.

**Stronger and better-characterized adversarial attacks.** The contractors had a tendency to produce adversarial examples that were relatively borderline or ambiguous, particularly when targeting more adversarially robust classifiers. However, when we attacked our models with our rewrite tool, we were able to construct more egregious adversarial examples, featuring direct injury, in part because researchers on our team used different heuristics for finding adversarial examples (see Appendix A.8). This underscores the need for a more diverse pool of stronger adversarial attacks, for better adversarial training [27]. Future work could add more tools (such as better suggestions for our human adversaries [50]) and study the relative effectiveness of the different tools, develop better training methods for human attackers, or more fully characterize properties of adversarial inputs to better understand our models [51, 52].

**Automated adversarial attacks with synthetic adversaries** In this work, we used human contractors (augmented with tools) to generate adversarial examples, as our task lacks an automated source of ground truth, we did not restrict our adversarial examples, and we were not successful in fine-tuning an LM adversary (as discussed in Appendix A.5). Future work could explore ways to generate synthetic examples, such as imitation learning on human examples [53] or better methods of using reinforcement learning to fine-tune automated adversaries [37].

**Exploring the generality of our results.** Much of our high level of reliability can be attributed to the fact that we were able to set particularly strict thresholds without significantly impacting quality on the story continuation task. Future work is needed to test whether or not this holds true on open-ended generation tasks in general.

**Adversarial training on larger models.** The classifiers we trained were 304M-parameter DeBERTa V3 models [44]. Most likely, many of their failures were due to capability limitations, and working with larger models would improve their performance substantially. On the other hand, we think that working with larger models would still leave us in qualitatively the same situation, since state-of-the-art models still fail to understand many things that humans do.

**Better techniques for measuring reliability.** Measuring the reliability of very robust classifiers by sampling randomly is very expensive. For example, on our test set of 100k examples, the difference between our best and worst classifiers was misclassifying 2 examples versus 3. Future work could attempt to use techniques similar to AMLS [54] to more precisely measure in-distribution and out-of-distribution reliability in an extremely-low-failure-rate setting, or define a upper bound on the reliability using techniques such as SDP relaxation [28].

## Acknowledgments and Disclosure of Funding

Paul Christiano originally proposed this project, and we benefited immensely throughout from discussions with him, as well as with Ajeya Cotra and Beth Barnes. We thank John Schulman, Jared Kaplan, Sam Bowman, Rohin Shah, Jonathan Uesato, Holden Karnofsky, Jan Leike, Jacob Hilton, Ethan Perez, Collin Burns, Jean-Stanislas Denain, Summer Yue, Nix Goldowsky-Dill, Chris MacLeod, Ryan Greenblatt, and Bill Zito for reading drafts of the paper and giving helpful feedback. We are grateful to Shauna Kravec, Dane Sherburn, and Everett Smith for their contributions to parts of the project, and to Kelsey Piper for organizing a party to collect more manual adversarial examples. We thank Surge and our contractors for their dedicated efforts over many months of labeling and writing adversarial examples. Finally, we thank the Redwood Research operations staff for providing an excellent work environment.

This work was funded by Redwood Research Group Inc.

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
