# OpenReview forum: "Adversarial training for high-stakes reliability"
_NeurIPS.cc/2022/Conference — NeurIPS 2022 Accept_

### Official Review · Reviewer_AYB8 · 2022-07-08

**Rating:** 9
**Confidence:** 4
**Soundness:** 4 excellent
**Presentation:** 4 excellent
**Contribution:** 4 excellent

**Summary:**

The authors aim to train a text classifier to classify injurious sentences. Their motivation involves understanding how to make systems that be reliably robust to failures in high-stakes settings such as ones that involve human injury. They use several strategies. After training on a labeled dataset, they manually generate adversaries, manually generate adversaries using feature attributions as an informative tool, and augment their adversaries with paraphrasing. They show that these strategies are useful and that performance on non-adversarial inputs was preserved.

**Questions:**

None that I’ll put here, but I have some mentioned below under Limitations.

**Limitations:**

There are more language data augmentation techniques than peraphrasing. Did the authors ever try any others? If so, mention this. If not, why not?

I mentioned above that I think feature attribution methods are overrated interpretability tools, that I struggle to think of other works that have used them for something genuinely useful, and that I am impressed by how this paper showed that they could be genuinely useful. But at the end of the day, I still believe that they are of limited usefulness, and that there are other interpretability techniques that have been used more successfully to identify adversarial weaknesses in networks. Having a human use feature attribution maps to generate adversaries is slow, relies on human creativity, and can, at best, only be used for local search. I would bet that synthesizing aversaries could be a better approach. See these three examples that come to mind for me (although the last one is from CV) [[https://arxiv.org/abs/2202.03286](https://arxiv.org/abs/2202.03286)] [[https://arxiv.org/abs/2005.00174](https://arxiv.org/abs/2005.00174) [[https://arxiv.org/abs/2110.03605](https://arxiv.org/abs/2110.03605)]. I also think that analyzing embeddings of examples to see which regions in embedding space could correspond to difficult examples may be very helpful for both interpreting and generating difficult examples. See [[https://arxiv.org/abs/2206.14754](https://arxiv.org/abs/2206.14754)] (although I understand that this paper only came out after the submission deadline). Finally, combining some sort of candidate adversary generation process with *weak* supervision from a human might make human oversight much more scalable. See [[https://arxiv.org/abs/2012.06046](https://arxiv.org/abs/2012.06046)].

So ultimately I think that the authors should think more about (1) different approaches for leveraging better interpretability tools, (2) synthesizing adversarial examples, and (3) better scaling the human oversight. At a minimum, I think that these should be discussed in future work. Arguably this paper is very simple because it adops some of the simplest and most brute force approaches possible for rigorous, high-stakes adversarial training, and I think that it will be important to expand on the simple methods presented here in future work. Perhaps its a good sign that this paper seem to suggest quite a bit more that could be done in the future to improve on things. I hope this gets accepted, and I encourage the authors to keep working in these 3 directions.

**Strengths And Weaknesses:**

I like how this work seems pointed toward reducing the risks from potentially highly intelligent systems that could make decisions themselves that directly affect human lives and welfare. While this isn’t a major concern right now, it might be later on. And even if I weren’t worried about the potential of future systems to potentially hurt people, this paper contains a number of tools and insights that seem valuable right now for training systems to have very low error.

I like how this paper uses interpretability tools to improve the process of generating adversaries for training. This proves the usefulness of the interpretability tools. They did a good deal of work to design and effectively use their human interface. And while this isn’t the most technical or mathy contribution, I don’t think that the value of designing an interface that is proven to help humans better work with these models should be understated. They don’t stress this too much, but I see a major part of the value of this paper as this human factors work. I generally think of feature attribution-based interpretability techniques to be bad because they don’t usually lead to useful, nontrivial, actionable insights about models. This paper doesn’t present itself as one on feature attribution-based interpretability, but they still do it very well. As someone who works in the area, I cannot think of any works involving feature attribution that have done something comparably useful. Perhaps this selling point of the paper could be discussed more.

I see the implications that this paper has for making genuinely safer AI systems to be very compelling and direct compared to almost all other papers in the field.

The paper is well written.

I think my main issues with this paper involve things that they didn’t try that I think they could and arguably should have. I’ll discuss these below under Limitations. That said, I feel very confident that this paper should clearly be accepted.

Minor

- The overlapping CI’s for table 3 seem to warrant a “somewhat” in line 250.
- You did not invent paraphrasing augmentation. Cite the original paper(s).
- I think the main paper should at least briefly mention false positives and refer to the appendix. I don’t think mention of comparing false positive rates should be completely omitted in the main paper.

---

> ### Author Response · Authors · 2022-08-02
> **Response to Reviewer AYB8**
>
> Thanks for the incredibly detailed and helpful response!
>
> > There are more language data augmentation techniques than peraphrasing. Did the authors ever try any others? If so, mention this. If not, why not?
>
> We did try a few other simple techniques like synonym substitutions, but preliminary experiments suggested that these techniques were more likely to change the meaning of the snippet while not really providing much diversity to the training set.
>
> We’ve updated the paper to cite paraphrasing-based augmentation, and have added some discussion of the false positive rate to the results section of the main paper.
>
> > So ultimately I think that the authors should think more about (1) different approaches for leveraging better interpretability tools, (2) synthesizing adversarial examples, and (3) better scaling the human oversight. At a minimum, I think that these should be discussed in future work.
>
> We agree that the three points you bring up ((1) additional interpretability tools (2) synthetic adversarial examples and (3) better oversight scaling) are interesting future directions to take this work. We’ve added some more discussion on these topics to the future work, and also added more discussion of synthetic adversaries to the methodology section.
>
> > See these three examples that come to mind for me (although the last one is from CV) [https://arxiv.org/abs/2202.03286] [https://arxiv.org/abs/2005.00174 [https://arxiv.org/abs/2110.03605]. I also think that analyzing embeddings of examples to see which regions in embedding space could correspond to difficult examples may be very helpful for both interpreting and generating difficult examples. See [https://arxiv.org/abs/2206.14754] (although I understand that this paper only came out after the submission deadline).
>
> We appreciated the pointers to other parts of the relevant literature, in particular Casper et al's work on feature-level adversaries from CV and Boecking et al’s work on weak supervision. We also agree that an important direction of future work is to fully categorize the failures of language models in the vein of Jain and Lawrence et al. We've added references to these work into our paper.

---

> > ### Comment · Reviewer_AYB8 · 2022-08-07
> > **Thanks for the response!**
> >
> > I have read the reviews and looked at updates to the paper. I think that the authors have considered and addressed some of the relatively minor suggestions that I pointed out. As far as my major concerns go, they continue to be things that this paper did not do as I discussed in my main review. But I also continue to be happy with the idea of leaving these for future work. I continue to see see this paper as high quality and with the potential for "groundbreaking impact".

---

### Official Review · Reviewer_x7BC · 2022-07-10

**Rating:** 5
**Confidence:** 4
**Soundness:** 3 good
**Presentation:** 4 excellent
**Contribution:** 2 fair

**Summary:**

In this paper authors propose to improve the binary classifier's performance by mining the hard minority class data (which authors call “adversarial examples”) by humans, a tool, humans augmented with the tool. The paper explains in detail how the whole data collection is done. The authors also train the model multiple times with new sets of adversarial examples found using the previous iteration of the model and show it gets harder to find hard examples automatically using the tool after a few iterations.

**Questions:**

-  If I understand correctly, the model is trained on balanced data. Have you tried any imbalanced cases too? How was the performance there?
- The problem the authors are addressing in this paper is somewhat related long-tail class distribution problem, I wonder if this proposed methodology can help there. Have you explored any multi-class setting?

**Strengths And Weaknesses:**

Strengths:
- Well written paper and easy to follow
- The authors describe in detail how they set up a system to mine the ambiguous negatives and use them to improve the robustness of the model.
- The investigations seem thorough.
- The use case of tools assisting humans is most interesting. I think that’s where the maximal benefit of the tool lies.


Weaknesses:
- The examples mined in this paper are not the typical adversarial examples we see in ML lingo, where we optimize a model to find an adversarial example. In the truest sense, the data the authors collect is “hard data” which the baseline model finds hard to classify.
- This is more of how they called data and training the model on ambiguous data improved the generalization of the model, which is not a surprising claim
- L266, Sec 5.4, there is previous work[1] suggesting that training models on hard/ambiguous data can improve the model’s generalizability.


[1] - Swayamdipta, Swabha, et al. "Dataset cartography: Mapping and diagnosing datasets with training dynamics." arXiv preprint arXiv:2009.10795 (2020).

---

> ### Author Response · Authors · 2022-08-02
> **Response to Reviewer x7BC**
>
> Thanks for taking the time to review our work! We'd like to respond to some of the points and questions raised.
>
> > The examples mined in this paper are not the typical adversarial examples we see in ML lingo, where we optimize a model to find an adversarial example. In the truest sense, the data the authors collect is “hard data” which the baseline model finds hard to classify.
>
> It's not clear to us what distinction is being drawn here. Like traditional adversarial attacks, our tool-assisted attack does iteratively optimize the input to achieve an increasingly egregious misclassification.
>
> > This is more of how they called data and training the model on ambiguous data improved the generalization of the model, which is not a surprising claim
> > L266, Sec 5.4, there is previous work[1] suggesting that training models on hard/ambiguous data can improve the model’s generalizability.
>
> Thanks for this citation. We didn’t observe this effect in our work - that is, training our models on adversarial data did not affect the performance on the in-distribution test set. However, this may be due to the difficulty of precisely measuring extremely low failure rates.
>
> ### Response to Questions:
> > If I understand correctly, the model is trained on balanced data. Have you tried any imbalanced cases too? How was the performance there?
>
> We performed a preliminary experiment which found that upsampling injurious snippets (that is, balancing the dataset) improved performance, though this effect did not reach statistical significance. We’ll add a note of this to the paper.
>
> > The problem the authors are addressing in this paper is somewhat related long-tail class distribution problem, I wonder if this proposed methodology can help there. Have you explored any multi-class setting?
>
> We do think that adversarial training could be a promising approach to address extreme class imbalance in some cases. We haven’t had the chance to apply adversarial training to such settings to quantify how much it would help.

---

### Official Review · Reviewer_zheV · 2022-07-18

**Rating:** 6
**Confidence:** 4
**Soundness:** 3 good
**Presentation:** 3 good
**Contribution:** 2 fair

**Summary:**

This work discusses the high-stakes settings where an AI model should avoid every instance of catastrophic failure. The authors specifically evaluate their work on a safe language generation task. They propose a toolkit for training networks using human adversaries and adversarial training. The authors claim that their models achieve very low false negative rates (FNR) without hurting the network's output quality. Further, their +tool-assisted model almost doubled the time it takes for a human contractor to find an adversarial example. This suggests that their trained classifiers are more robust.

**Questions:**

1. The authors claim that they allow examples in unrestricted space in the related works section. However, the human contractors are given recommendations for tokens from deberta-v2-xlarge. This restricts the adversaries from not creating smart examples using *hidden vocabulary* [1]. Does this mean that the adversaries are created in a restricted space?
2. Table 2 shows that the FNR for baseline and all the other trained classifiers are almost the same. How is the proposed method better than the baseline using this FNR metric?
3. The FNR is evaluated on a non-adversarial test dataset. How does this serve to be a good metric to evaluate catastrophic failure? Why is the FNR not evaluated on an adversarial test dataset?
4. How would the proposed method perform without the +manual step in the pipeline? Removing human involvement can improve the scalability of the pipeline. How does that affect the network robustness?


Reference:

[1] Daras, Giannis, and Alexandros G. Dimakis. "Discovering the Hidden Vocabulary of DALLE-2." arXiv preprint arXiv:2206.00169 (2022).

**Limitations:**

Yes, the paper discusses its limitations regarding stronger adversarial attacks, the generality of results, lack of theory, and a need for better metrics to measure reliability.

**Strengths And Weaknesses:**

Strengths:
+ The paper aims toward building reliable AI systems that achieve better worst-case performance.
+ The authors provide detailed supplementary material describing the implementation details.
+ A novel toolkit is designed for adversarially training networks using human and automated adversaries.
+ A smart toolkit to help human contractors find adversarial examples using saliency mapping and token recommendation.
+ The final trained classifier achieves very low false negative rates $-$ 2/2447.
+ The time required by human contractors to find adversarial examples (for the final trained classifier) doubles. This suggests that the proposed adversarial training technique improves robustness.

Weaknesses:
- The problem statement highlights worst-case reliability (even a single catastrophic failure should not ideally occur). I think it's hard to make claims without any theoretical evidence. Also, I think the toolkit should be more rigorously analyzed to show more empirical evidence (eg, on image datasets).
- The toolkit relies on human adversaries; hence, scaling the framework for other tasks could be laborious.
- Though the proposed adversarial training technique makes it harder for a human contractor to find an adversarial example, it's still not impossible for a human adversary to attack the final trained classifiers.
- Figure 3 is not referred to in any part of the main text. FNR on snippets from the tool-assisted attack on the classifiers trained using the proposed technique is quite high. This doesn't satisfy the demands of the problem statement.
- In appendix A.3, the authors mention that the data from the validation set made it into the training set due to a bug in the tool. This affects hyperparameter selection. I would appreciate it if the authors provided results after fixing this bug.
- The proposed technique is not compared against other obvious baselines (eg, constrained Lp-norm based adversarial training).
- The supplementary material does not contain codes for checking the experiment implementations.

---

> ### Author Response · Authors · 2022-08-02
> **Response to Reviewer zheV**
>
> Thanks for the detailed response!
>
> > Also, I think the toolkit should be more rigorously analyzed to show more empirical evidence (eg, on image datasets).
>
> We agree that exploring adversarial training’s ability to achieve high-stakes reliability in many domains (such as images) would be interesting. However, the vast majority of adversarial training papers focus on a few tasks in single domain due to the effort required to perform adversarial attacks (for example, this previous NeurIPS adversarial training paper only study CIFAR-10 and Image Net: [1], while this one reports on only results from CIFAR-10 in the main text: [2]) and the fact that most adversarial attacks are domain specific (for example, many gradient-based attacks don’t work in discrete domains such as language). In our specific case, replicating our attacks on a different domain would’ve required developing new domain-specific infrastructure as well as retraining our team of contractors.
>
> Moreover, almost all papers on adversarial robustness focus exclusively on restricted adversarial examples on the vision domain, so the focus on natural language in this work increases the diversity of the diversity of the literature (we are only aware of a few exceptions that study unrestricted adversarial examples in language, such as [3], [4], and [5])
>
> > The proposed technique is not compared against other obvious baselines (eg, constrained Lp-norm based adversarial training).
>
> An Lp-norm based adversarial attack doesn’t make sense in the NLP setting we consider. Unlike with images, small perturbations in a sentence can change the meaning of the sentence drastically.
>
> > The problem statement highlights worst-case reliability (even a single catastrophic failure should not ideally occur). I think it's hard to make claims without any theoretical evidence.
>
> We agree that it’s hard to claim to have solved worst-case reliability without a formal proof. However, we make no such claim. As stated on line 59-61 of the introduction, we see our work as a first step toward solving the problem we proposed in the paper. As the reviewer notes elsewhere, the adversarial training techniques used in this work do not successfully eliminate all of the failures that we can find with existing techniques. To us, this suggests that there’s still significant value in empirical work in this domain.
>
> > The toolkit relies on human adversaries; hence, scaling the framework for other tasks could be laborious.
>
> We agree that scaling human adversaries to other domains will require effort, but we don’t see this as a particular limitation of our work. As we wish to study adversarial robustness in an unrestricted setting without an automated source of ground truth, our adversarial training process will always require some amount of human judgment. We’ll add some discussion of this to the paper.
>
> > Figure 3 is not referred to in any part of the main text. FNR on snippets from the tool-assisted attack on the classifiers trained using the proposed technique is quite high. This doesn't satisfy the demands of the problem statement.
>
> This is incorrect: we discuss figure 3 in Section 5.3, in the paragraph starting “Classifiers trained against more adversarial attacks were more robust to other adversarial attacks.”:
> “We treated each set of tool-assisted rewrites as a separate dataset and evaluated each classifier against each dataset. Figure 3 shows the results.”
> We agree that adversarial training was not able to fully harden a classifier against our tool-assisted attack. We don’t claim in our paper that we’ve solved the problem of high-stakes reliability; again, we see the contributions of our work as outlining the problem and proposing a novel tool-assisted adversarial attack.
>
> > In appendix A.3, the authors mention that the data from the validation set made it into the training set due to a bug in the tool.
>
> As we had over 100k snippets in the validation set and only 15 snippets from the validation set were accidentally included in the training set, this actually didn’t change which hyperparameters were chosen. We’ll clarify this in the appendix.
>
> ### References
>
> [1] Pang, Tianyu, et al. "Boosting adversarial training with hypersphere embedding." Advances in Neural Information Processing Systems 33 (2020): 7779-7792.
>
> [2] Liu, Feng, et al. "Probabilistic margins for instance reweighting in adversarial training." Advances in Neural Information Processing Systems 34 (2021): 23258-23269.
>
> [3] Wallace, Eric, et al. "Trick me if you can: Human-in-the-loop generation of adversarial examples for question answering." Transactions of the Association for Computational Linguistics 7 (2019): 387-401.
>
> [4] Wallace, Eric, et al. "Analyzing dynamic adversarial training data in the limit." arXiv preprint arXiv:2110.08514 (2021).
>
> [5] Perez, Ethan, et al. "Red teaming language models with language models." arXiv preprint arXiv:2202.03286 (2022).

---

> > ### Author Response · Authors · 2022-08-02
> > **Answers to questions raised by Reviewer zheV**
> >
> > > The authors claim that they allow examples in unrestricted space in the related works section. However, the human contractors are given recommendations for tokens from deberta-v2-xlarge. This restricts the adversaries from not creating smart examples using hidden vocabulary [1]. Does this mean that the adversaries are created in a restricted space?
> >
> > Our tool allowed our human adversaries to enter arbitrary text strings, regardless of the recommendations of our snippet rewrite tool. We’ll update the description of our tool in the paper to clarify this point. We agree that our human adversaries were biased toward tokens that are high probability according to deberta-v2. Given the purpose of the classifier is to filter a generator, we believe this is reasonable.
> >
> > > Table 2 shows that the FNR for baseline and all the other trained classifiers are almost the same. How is the proposed method better than the baseline using this FNR metric?
> >
> > As stated in the abstract and the intro and discussed in section 5.4, performing adversarial training did not affect the FNR on in-distribution data. However, the adversarially trained models were better according to other metrics, such as the difficulty of finding adversarial examples, or the FNR on adversarially generated datasets (section 5.3).
> >
> > > The FNR is evaluated on a non-adversarial test dataset. How does this serve to be a good metric to evaluate catastrophic failure? Why is the FNR not evaluated on an adversarial test dataset?’
> >
> > We did evaluate the FNR on adversarial test datasets (Figure 3, discussed in section 5.3). In addition, we’ve updated the paper to include the FNR metrics on all of the datasets in the appendix. The reason we included FNR on the in-distribution dataset in section 5.4 is to answer the question, “does adversarial training on out-of-distribution (adversarial) data affect in-distribution performance?”. As we state in section 5.4, we don’t see an effect (either positive or negative) on in-distribution performance.
> >
> > > How would the proposed method perform without the +manual step in the pipeline? Removing human involvement can improve the scalability of the pipeline.
> >
> > As previously mentioned, due to a lack of automated ground truth, some amount of human involvement is required in our training pipeline.

---

> > > ### Comment · Reviewer_zheV · 2022-08-05
> > > **Thank you for the detailed response**
> > >
> > > Dear authors,
> > >
> > > My concerns are well addressed. Hence, I am increasing my score. Thanks for the detailed response

---

### Author Response · Authors · 2022-08-02
**Brief response to reviews**

We’d like to thank the reviewers for their detailed reviews. We’re happy that the reviewers agree that the problem of high-stakes reliability is important, that our paper is generally well written (reviewers x7BC and AYB8) and that our token-substitution tool is interesting and novel. We also appreciate the suggestions on clarity and related work, and have updated our paper based on feedback.

---

### Meta-Review · Area_Chair_bgYk · 2022-08-23

**Recommendation:** Accept
**Confidence:** Certain

**Metareview:**

The paper used a safe language generation task (``avoid injuries'') as a testbed for achieving high reliability through adversarial training. Reviewers had a high variance in their evaluation of this work. While reviewers found the toolkit interesting and the paper well-written, there were some concerns regarding the lack of theoretical evidence on worst-case reliability and the lack of strong/adaptive attacks and baselines in experiments. Given all, I think the paper is above the accept threshold.

**Award:**

No

---

### Decision · Program_Chairs · 2022-09-14

Accept